# Probabilistic Guarantees for Abductive Inference

## Abstract

Abductive reasoning is ubiquitous in artificial intelligence and everyday thinking. However, formal theories that provide probabilistic guarantees for abductive inference are lacking. We present a quantitative formalization of abductive logic that combines Bayesian probability with the interpretation of abduction as a search process within the Algorithmic Search Framework (ASF). By incorporating uncertainty in background knowledge, we establish two novel sets of probabilistic bounds on the success of abduction when (1) selecting the *single* most likely cause while assuming noiseless observations, and (2) selecting *any* cause above some probability threshold while accounting for noisy observations. To our knowledge, no existing abductive or general inference bounds account for noisy observations. Furthermore, while most existing abductive frameworks assume exact underlying prior and likelihood distributions, we assume only percentile-based confidence intervals for such values. These milder assumptions result in greater flexibility and applicability of our framework. We also explore additional information-theoretic results from the ASF and provide mathematical justifications for everyday abductive intuitions.

## 1 Introduction

Imagine a patient visits a doctor because of a persistent cough, fever, and shortness of breath. As the doctor considers these symptoms and the prevalence of certain illnesses in the area, the doctor may hypothesize that the patient has pneumonia. This is an example of abductive reasoning, or *abduction.*

Abduction is the process of finding the best causal explanation given some observed effects. Abductive reasoning can be categorized into strategies that can generate new hypotheses, known as *creative abduction*, and those that select the best candidate given a set of possible explanations, known as *selective abduction* (Schurz, 2007). We focus on selective abduction, which can be formalized with Bayesian Decision Theory (Romeijn, 2013). Given observation(s) $O$, we select a hypothesis $C_i$ from a finite set of hypotheses $C$. Per Bayesian probability, we denote $\Pr(C_i|O)$ as the *posterior*, where the most probable cause is that with the highest posterior. By Bayes' theorem,

$$\Pr(C_i|O) = \frac{\Pr(O|C_i)\Pr(C_i)}{\Pr(O)}.$$

However, during the hypothesis selection process, the relevant observations $\Pr(O)$ remain constant. Thus, the relevant form of Bayes' theorem becomes

$$\Pr(C_i|O) \propto \Pr(O|C_i)\Pr(C_i).$$

To perform selective abduction, one simply chooses the hypothesis whose likelihood and prior have the greatest product.[1]

Abduction accompanies induction and deduction as one of three forms of logical reasoning (Rodrigues, 2011; Peirce et al., 2017). In supervised machine learning, inductive and abductive processes serve as the

---

[1]Note that a general cause (i.e., one that is likely and can produce many effects) is not guaranteed to have a high posterior since its smaller likelihood counters the effect of its high prior.

underlying logic behind model training and application (see Figure 1) (Mooney, 2000). While both inductive and abductive reasoning are applied ubiquitously in the field, inductive reasoning is currently the more well-understood process; we have already gained a theoretical understanding of inductive accuracy (Dietterich, 1989; Kietz, 1993; Cummings et al., 2016; Garg et al., 2021; Cosentino et al., 2022). However, to our knowledge, there currently exist no formal quantitative frameworks with accuracy bounds for abductive reasoning.

In a broader context, artificial intelligence researchers such as Erik Larson argue that obtaining a quantitative theory of abduction is a necessary step towards bridging machine and human intelligence. Abduction, more specifically creative abduction, encapsulates human intuition or "guessing" capability lacking in current models. Larson describes machine understanding of abductive reasoning as the central "blind spot" of artificial intelligence:

> *"Abductive inference is required for general intelligence, purely inductively inspired techniques like machine learning remain inadequate…The field requires a fundamental theory of abduction." (Larson, 2021)*

Our work primarily aims to (1) provide currently lacking accuracy bounds for abductive reasoning and (2) serve as a preliminary version of this "fundamental theory of abduction" needed for abductive machine understanding. We propose a general probabilistic framework for *selective* abduction built from Bayesian Decision Theory (Berger, 2013) (detailed in Section 3), serving as a jumping off point for future work on creative abduction. Through this Bayesian framework, we first derive upper and lower probabilistic bounds of abductive accuracy when assuming underlying $q$-percentile uncertainty bounds of prior and likelihood probabilities for each cause (Section 4). This first set of accuracy bounds treats successful abduction as choosing the *single* true hypothesis assuming the selection of the single highest posterior. We then extend this by reframing abduction as a search process within the Algorithmic Search Framework (ASF) (Montañez, 2017), which lets us describe and bound the probability of selecting *any* hypothesis with a posterior probability above a certain threshold while accounting for noisy observations (Section 5.1). Lastly, in addition to deriving bounds on abductive accuracy, we apply the framework to quantitatively justify common-sense heuristic abduction (Section 5.2, 5.3).

## 2 Related work

We review applications of abductive logic in machine learning and artificial intelligence, and survey existing abductive frameworks and current literature on Bayesian inference.

### 2.1 Logic in Machine Learning

Peirce introduced abduction alongside induction and deduction as the three pillars of logical inference (Shanahan, 1986). Induction, inferring causal relationships from data, is central to machine learning (Mooney, 2000). Inductive logic is core to the training process, where labeled examples are used to develop generalized relationships within a model. Deductive and abductive logic are employed within machine learning's underlying inductive framework by applying the relationships derived through inductive training (Bergadano et al., 2000). Deduction facilitates data generation by selecting a class (cause) to produce feature data (observations). Conversely, abduction involves assigning class labels (causes) to unlabeled data (observations) using a trained model that embeds established causal relationships (see Figure 1) (Bergadano et al., 2000).

Induction corresponds to the training phase, where input-output relationships are learned, while abduction relates to classification, using known relationships to infer likely causes. Table 1 outlines the connections between logical inference (Bergadano et al., 2000) and machine learning.

From this perspective, machine learning applies abductive logic in model inference. For example, machine learning emulates the abductive reasoning used in spam detection and medical diagnosis by applying trained algorithms to unlabeled data (i.e., text from emails or radiology scans). However, model inference is just one of many applications of abduction in machine learning. Our work addresses the theoretical limits of the success of abductive reasoning generalizable to applications such as these.



Figure 1: Three methods of inference. The dotted lines show which part of each process is being inferred.

Table 1: Schematic outline of the processes of inference in supervised machine learning (Bergadano et al., 2000).

| Logical Inference | Machine Learning |
|---|---|
| Induction:
$P(a)$
$\therefore \forall w P(w)$ | Training:
$(x^1, y_1), ..., (x^n, y_n)$
$\therefore f : \mathcal{X} \to \mathcal{Y}$ |
| Abduction:
$Q(a)$
$P(w) \to Q(w)$
$\therefore P(a)$ | Classification:
$\mathbf{x}^m$
$y_m = f(\mathbf{x}^m)$
$\therefore y_m$ |

## 2.2 Applying Abduction in Machine Learning

In addition to its synonymy with the higher-level logic of model inference, abductive logic is central to several common machine-learning processes.

Abduction is the underlying logic of Bayesian networks, which are used for tasks such as clustering, supervised classification, anomaly detection, and temporal modeling (Mihaljević et al., 2021). Bayesian networks are particularly useful for decision-making under uncertainty (Mihaljević et al., 2021) and are widely used in criminology and prognosis, diagnosis, and prescription in healthcare (Song et al., 2021).

Additionally, *maximum a posteriori* (MAP) applies abductive reasoning through Bayes' theorem to optimize model parameters.[2] Analogizing training data $D$ as observations and a possible model parameterization $\mathbf{w}$ to a possible cause, MAP optimizes parameters by maximizing the posterior, $\Pr(\mathbf{w}|D)$ (Bishop, 2006).

Abductive reasoning is also prevalent in relational learning and computer vision. In relational learning, where data is represented through relationships with other data, abduction guides search and generates missing input data (Bergadano et al., 2000). In computer vision, integrating abductive reasoning with convolutional neural networks (CNNs) enhances spatial-temporal reasoning and image segmentation tasks, which also contributes to explainable AI by incorporating understandable reasoning into black-box models (Zhang et al., 2021; Rafanelli et al., 2023).

## 2.3 Formalizations of Abduction

Various formalizations of abduction have been explored in symbolic AI literature (Paul, 2000). Set-cover-based approaches involve selecting a subset of hypotheses from a larger set, requiring complete causal relationships (Allemang et al., 1987). Knowledge-level approaches propose explanations based on beliefs (Levesque, 1989). Abductive Logic Programming (ALP) represents inferences as entailments from a prior knowledge base to the veracity of specific causes (Kakas et al., 1992; Alberti et al., 2008; Raghavan & Mooney, 2010).

---

[2]Note that training remains an inductive process on a larger scale; MAP applies abductive logic within training steps since it is a Bayesian method.

Probabilistic Horn Abduction extends Prolog by combining exact probabilities of hypotheses with Bayes' theorem to generate posterior probabilities built from multiple observations (Poole, 1991; Ng & Mooney, 1991). Unlike our proposed framework, it assumes exact prior and likelihood probabilities and does not incorporate confidence ranges for these distributions (Poole, 1991). Developments in probabilistic and probabilistic abductive logic programming (Turliuc et al., 2013; Azzolini et al., 2021) depart from our work in similar ways, as exact probabilities are assumed and general bounds for abductive success are not provided.

A recently developed framework applying stochastic mathematical systems (SMSs) models abduction by representing reasoning as stochastic systems, with the human reasoner SMS generating hypotheses and an oracle SMS evaluating their validity based on explanatory power and evidence (Wolpert & Kinney, 2024). Like Probabilistic Horn Abduction, it also does not account for the uncertainties in underlying distributions.

These methods lack probabilistic guarantees for the correctness of the abductive inferences and do not quantify associated uncertainties. Our approach addresses this gap by integrating formal machine learning frameworks, which allows for more precise quantification of the uncertainties involved in abductive inferences.

### 2.4 Bayesian Inference

Bayesian inference forms the basis of our framework, deriving accuracy bounds using $q_p$ and $q_l$ confidence intervals for prior and likelihood distributions, respectively. These intervals represent confidence in causal relationships ($q_l$) and general world knowledge ($q_p$), providing flexibility in representation.

Bayesian inference estimations and bounds are well-explored in the literature, with numerous known methods of deriving accuracy bounds for inference of specific algorithms or tasks (Yekutieli, 2012; Pati et al., 2018; Chérief-Abdellatif et al., 2019; Alroobaea et al., 2020; Audibert, 2009; Zhang et al., 2021; Alquier & Ridgway, 2020; Ferguson et al., 1992; Cox, 1993; Alvarez et al., 2014). However, general methods for deriving bounds using techniques like multi-valued mapping (Dempster, 1968) or prior measure intervals (Dempster, 1967) are less common. To our knowledge, no existing method derives Bayesian inference bounds based on specific prior and likelihood confidence intervals with probabilities $q_l$, as our framework does.

Our work is the first to leverage the ASF (Montañez, 2017) to construct a formalization of abduction or abduction by Bayesian inference. Unlike other established frameworks (Poole, 1991; Ng & Mooney, 1991; Poole, 1993), the ASF accounts for noisy observations – observations that may not fully reflect "true" events. The framework makes very few assumptions of given information resources, $F$, which (in the case of abduction) embeds observation data. Such data is abstracted as binary strings, with no conditions placed on what form the binary strings take, only that we have functions available to extract feedback from the strings for individual search queries. Thus, with no restrictions placed on the information resources, the ASF accommodates both noisy and noiseless observations. To our knowledge, there are no abductive or general inference bounds with this specific property. Existing work has only analyzed the correlation of real dataset noise with the accuracy of Bayesian inference for specific algorithms, assuming specific data qualities (An et al., 2012).

## 3 Preliminaries

We formalize the fundamental building blocks of abduction, causes and observations, as vectors. The vectorization of such outcomes allows the formalization of posterior, likelihood, and prior probabilities as distributions over a vector space. We then formalize the likelihood and posterior uncertainty intervals on which the abductive search process relies.

### 3.1 Vectorizing Observations

We formalize observations as binary vectors, where each scalar component corresponds to the existence of a specific *observation feature* or certain observed outcome. For example, suppose you swallow an unknown pill and then your headache disappears. A representative observation vector might be $\langle 1, 1 \rangle$ with each feature representing (1) "Did you swallow a pill?" and (2) "Did the headache go away?" (respectively). If the headache disappeared without taking a pill, the observation vector would be $\langle 0, 1 \rangle$.

**Definition 3.1.** ($\mathcal{O}$) Let $\mathcal{O}$ denote the vector space of discrete topology containing all binary-featured observation vectors.

Since any outcome must strictly occur or not occur, the set of possibilities within $\mathcal{O}$ is mutually exclusive and collectively exhaustive. In the case where an observation is a continuous variable, such as temperature, we would convert the variable by adding additional features representing levels of the value, such as ["cold", "lukewarm","hot"].[3]

**Proposition 3.1.** *The set of possible outcomes represented as vectors within $\mathcal{O}$ is mutually exclusive and collectively exhaustive.*

### 3.2 Vectorizing Causes and Likelihood Probability Mass Functions

A cause $C_i$ has some probability of instigating any possible observation vector $\mathbf{x} \in \mathcal{O}$, inducing a conditional probability mass distribution (i.e., likelihood function) $\Pr(\mathbf{x}|C_i)$ over all observations $\mathbf{x} \in \mathcal{O}$. Note that every observation $\mathbf{x} \in \mathcal{O}$ is disjoint (Proposition 3.1), and we assume exactly one observation vector is produced and observed.

Following the earlier example, the likelihood distribution over the observation space for the cause "aspirin" expresses the probability that, *assuming* aspirin was taken, phenomena $\mathbf{x} \in \mathcal{O}$ would follow. Knowing that aspirin typically relieves headaches and is ingested in pill form, the likelihood distribution over $\mathcal{O}$ with dimensions {"Pill taken?", "Headache relieved?"} may be similar to Table 2. Such a likelihood distribution depends only on the cause $C_i$, and will act over $\mathcal{O}$. The notation $do(\text{aspirin} = \texttt{True}))$ denotes that we taking some action to force the condition "aspirin" to be $\texttt{True}$, per do-calculus (Pearl et al., 2000).

Table 2: Example likelihood distribution for effects of aspirin.

| Pill taken? | Headache relieved? | $\mathbf{x}$ | $\Pr(\mathbf{x}|do(\text{aspirin} = \texttt{True}))$ |
|:---:|:---:|:---:|:---:|
| no | no | $\langle 0, 0 \rangle$ | 0.05 |
| no | yes | $\langle 0, 1 \rangle$ | 0.10 |
| yes | no | $\langle 1, 0 \rangle$ | 0.15 |
| yes | yes | $\langle 1, 1 \rangle$ | 0.70 |

Assuming that exactly one of the observation vectors must occur, we know that the probabilities for each collectively must sum to one. Considering all the possible ways there are to assign probabilities to a collectively exhaustive and mutually exclusive set of options forms a mathematical simplex, $\mathcal{S}$. For $k$ observation features (where $\dim(\mathcal{O}) = k$), simplex $\mathcal{S}$ forms a continuous $2^k - 1$ dimensional hyperplane containing all possible "cause vectors", each corresponding with some likelihood probability mass function over the $2^k$ observation vectors in $\mathcal{O}$. Each scalar component of a $2^k$ dimensional "cause" vector $\mathbf{c} \in \mathcal{S}$ denotes how much probability mass is placed on a corresponding observation vector in $\mathcal{O}$. Since we define a "cause" as the event representation of a likelihood distribution over $\mathcal{O}$, a single cause vector in $\mathcal{O}$ can actually represent multiple concurrent events or causes.

Ensuring that every cause $\mathbf{c} \in \mathcal{S}$ corresponds to a valid probability mass function on $\mathcal{O}$ requires the following two properties: (1) the simplex is bounded within $[0, 1]$ on every dimension such that no $\mathbf{c} \in \mathcal{S}$ holds a component that indicates an invalid probability, and (2) the sum of all components of a cause vector equals 1.

### 3.3 Defining Posterior Confidence Bounds

During the decision-making process, we compare different posterior probabilities for the same observation $\mathbf{x}$. Since the evidence, $\Pr(\mathbf{x})$, is constant, we will only compare the product of the likelihood and prior across causes, namely, $\Pr(\mathbf{x}|\mathbf{c})\Pr(\mathbf{c})$, which we will be referring to as the "posterior" for simplicity.

In life, we often lack these exact likelihood and prior distributions. Instead, we may estimate such probabilities through numerical techniques, including asymptotic estimations, Monte Carlo methods, numerical integration, and various sampling methods (Tierney, 1994; Chib, 1996; Levine & Casella, 2001). Other distribution

---

[3]Note that any probability mass function over $\mathcal{O}$ would, by default, place zero mass on contradictory observation vectors, such as one that is both "hot" and "cold."

estimation methods include smoothing and reduction methods, and Markov chain algorithms can be further used to combine estimation methods (Tierney, 1994). Thus, to account for uncertainty, we estimate likelihood, prior, and posterior probabilities through confidence intervals.

We define two functions denoting the upper bound likelihood probability, $l_U(\mathbf{c}, \mathbf{x})$ , and lower bound likelihood, $l_L(\mathbf{c}, \mathbf{x})$, of the $q_l$-percentile likelihood uncertainty interval, where $l_U(\mathbf{c}, \mathbf{x}) \geq l_L(\mathbf{c}, \mathbf{x})$. The prior $q_r$-percentile uncertainty interval is similarly represented through an upper and lower bound $r_U(\mathbf{c})$ and $r_L(\mathbf{c})$ (respectively).

The upper and lower confidence bounds of the posterior, $p_U(\mathbf{c}, \mathbf{x})$ and $p_L(\mathbf{c}, \mathbf{x})$, can then be found by simply multiplying the upper or lower bounds of the likelihood and prior probabilities together:

$$p_U(\mathbf{c}, \mathbf{x}) = l_U(\mathbf{c}, \mathbf{x}) r_U(\mathbf{c}) \quad \text{and} \quad p_L(\mathbf{c}, \mathbf{x}) = l_L(\mathbf{c}, \mathbf{x}) r_L(\mathbf{c}).$$

This bound assumes there is a $q_l$ probability that the likelihood lies in its $q_l$-percentile interval $[l_L(\mathbf{c}, \mathbf{x}), l_U(\mathbf{c}, \mathbf{x})]$ and, likewise, that there is a $q_r$ probability that the prior lies in its $q_r$-percentile interval $[r_L(\mathbf{c}, \mathbf{x}), r_U(\mathbf{c}, \mathbf{x})]$. Thus, the interval $[p_U(\mathbf{c}, \mathbf{x}), p_L(\mathbf{c}, \mathbf{x})]$ defines the $q$-percentile confidence interval for posterior $\Pr(\mathbf{c}|\mathbf{x})$ where $q = q_l q_r$.

### 3.4 Narrowing the Space of Possible Causes

We have established $\mathcal{S}$ as the *infinite* space containing all possible likelihood distributions over $\mathcal{O}$ and, thus, the space of *all* possible causes. However, this space includes likelihood distributions generated by causes that are implausible. In the real world, we often choose the most likely cause from a smaller set of plausible causes; for example, one would not consider an atomic bomb to be a plausible cause for your headache disappearing. Rather than considering the entirety of $\mathcal{S}$ as the pool of possible causes, we assume that some finite subset $\mathcal{C} \subset \mathcal{S}$ with cardinality $k = |\mathcal{C}|$ has been pre-selected as the finite set of *plausible* causes assumed to contain the true cause. We further assume $\mathcal{C}$ includes a "cause" $C_{\text{other}}$, whose posterior encapsulates the (likely low) combined probability of all other causes in $\mathcal{S}$ occurring. With this, we assume that all causes in $\mathcal{C}$ are disjoint and that $\mathcal{C}$ contains the *one* true explanation for observation $\mathbf{x}$ (namely, what actually caused it).

**Definition 3.2.** ($\mathcal{C}$) Let $\mathcal{C} \subset \mathcal{S}$ denote the relevant finite subset of possible cause vectors in $\mathcal{S}$.

For notational simplicity, we additionally denote each cause as $C_i \in \mathcal{C}$ and its corresponding "true" posterior probability as $M_i$ in posterior set $\mathcal{M}$. We likewise simplify the notation of the upper and lower bounds of $q$-percentile uncertainty interval posterior $M_i$ as follows: from $p_U(\mathbf{c}, \mathbf{x})$ and $p_L(\mathbf{c}, \mathbf{x})$ to $u_i$ and $l_i$, respectively. For future reference, we define the following:

**Definition 3.3.** ($M_i$) Let $M_i \in \mathcal{M}$ denote the "true" posterior probability of cause $C_i \in \mathcal{C}$, where $M_i = \Pr(C_i|\mathbf{x}) \Pr(C_i)$. Then $M_i$ falls into the following uncertainty interval with probability $q$:

$$M_i \in [l_i, u_i].$$

Since we assume each $C_i \in \mathcal{C}$ is disjoint, and that $\mathcal{C}$ surely contains the true explanation for observation $\mathbf{x}$, each posterior probability $\Pr(C_i|\mathbf{x})$ sums to 1. Thus,

$$\sum_{M_i \in \mathcal{M}} M_i = \Pr(\mathbf{x}).$$

**Definition 3.4.** ($\mathcal{U}$) Let $\mathcal{U}$ denote the set containing the $q$-percentile uncertainty interval bounds $[l_i, u_i]$ for each posterior $M_i \in \mathcal{M}$.

Note that we also assume $|\mathcal{M}| = |\mathcal{C}| = |\mathcal{U}| \geq 2$, as determining the most likely cause from a set of only one is trivial.

## 4 Abduction by Bayesian Inference

### 4.1 Cause Selection with Uncertainty Intervals

Given the set of $q$-percentile confidence posterior probability uncertainty bounds $[l_i, u_i] \in \mathcal{U}$ for each cause $C_i \in \mathcal{C}$, one selects the cause whose *point estimate posterior probability* is highest. Since the true posterior

probabilities of each cause are unknown, this process may incorrectly select a cause whose posterior is not the true maximum. We quantify this rate of incorrect selection in the case where every posterior $M_i \in \mathcal{M}$ is contained in respective confidence bound $[l_i, u_i]$. Let predicate IsMax($M_i$) denote whether posterior $M_i$ is truly the highest posterior. We first define the probability range where the maximum posterior must lie, $[l, u]$.

**Definition 4.1.** Let each posterior $M_i \in \mathcal{M}$ occur within $q$-percentile confidence interval $[l_i, u_i] \in \mathcal{U}$. Then, we set

$$l = \max(\{l_i | i \in \mathbb{Z}_+, i \leq |M|\})$$

and

$$u = \max(\{u_i | i \in \mathbb{Z}_+, i \leq |M|\}).$$

**Proposition 4.1.** *Assuming that every $M_i \in \mathcal{M}$ lies in respective $q$-percentile confidence interval $[l_i, u_i] \in \mathcal{U}$, the max posterior is bounded by $u$ and $l$.*

Thus, in the case that *every* confidence bound fully contains its respective posterior almost surely (instead of just with probability $q$), any posterior $M_i$ whose uncertainty bounds $[l_i, u_i]$ overlap with $[l, u]$ is potentially the maximum posterior with some probability $\Pr(\text{IsMax}(M_i))$.

**Theorem 4.2.** *Let $\mathcal{M}' \subseteq \mathcal{M}$ denote the set of posteriors whose confidence intervals intersect with $[l, u]$. The probability that $M_i \in \mathcal{M}'$ is the maximum posterior is as follows:*

$$\Pr(\text{IsMax}(M_i)) = \int_l^u P\left(M_i = x, \bigcap_{\substack{M_j \in \mathcal{M}', \\ C_j \neq C_i}} (M_j < x)\right) dx.$$

This accounts for any estimated posterior probability distribution within $[l_i, u_i]$, but assumes $M_i$ is contained by $[l_i, u_i]$ with probability 1.[4]

### 4.2 Bayes Error Rate

However, even assuming the cause with the true highest posterior is successfully identified, there is the unavoidable error from non-zero posteriors of the "losing" categories. The true cause of a feature may simply not have the highest posterior. This minimum achievable error is expressed by Bayes Error Rate (BER):

**Definition 4.3.** ($\epsilon$, (Sekeh et al., 2020)) Let $\epsilon$ denote Bayes multiclass error rate (BER) for every $C_i \in \mathcal{C}$. For $|\mathcal{C}| = k$ possible causes:

$$\epsilon = 1 - \int \Pr(\mathbf{x}) \max_i \Pr(C_i | \mathbf{x}) d\mathbf{x}.$$

However, the formula above is often impractical to compute for $k > 2$ causes. Instead, one can derive bounds for the multi-cause BER with techniques such as the Bhattacharyya bound, estimations using Friedman-Rafsky test statistics, and non-parametric bounds using Henze-Penrose divergence (Sekeh et al., 2018). We adopt a recent method[5] of upper bounding BER through global minimal spanning trees (Sekeh et al., 2020) and adopt a pairwise computational lower bounding method for BER (Lin, 1991).

**Definition 4.4.** ($\epsilon_{\text{upper}}$, (Sekeh et al., 2020)) Let $\epsilon_{\text{upper}}$ denote the upper bound of BER such that $\epsilon \leq \epsilon_{\text{upper}}$. Then, for $|\mathcal{C}| = k$,

$$\epsilon_{\text{upper}} = 2 \sum_{i=1}^{k-1} \sum_{j=i+1}^{k} \delta_{ij}$$

where

$$\delta_{ij} := \int \frac{\Pr(C_i) \Pr(C_j) \Pr(\mathbf{x}|C_i) \Pr(\mathbf{x}|C_j)}{\Pr(C_i) \Pr(\mathbf{x}|C_i) + \Pr(C_j) \Pr(\mathbf{x}|C_j)} d\mathbf{x}.$$

---

[4]See the appendix for directly computable forms of Theorem 4.2.
[5]This method provides a tighter bound than aforementioned techniques (Sekeh et al., 2020).

**Definition 4.5.** ($\epsilon_{\text{lower}}$, (Wisler et al., 2016), (Lin, 1991)) Let $\epsilon_{\text{lower}}$ denote the lower bound of BER such that $\epsilon \geq \epsilon_{\text{lower}}$. BER may be lower bounded by applying pairwise computations of Bayes error $\epsilon_{ij}$ for $i$ and $j$ between every unique cause pair $(C_i, C_j)$ where $C_i \in \mathcal{C}, C_i \in \mathcal{C}, i \neq j$:

$$\epsilon_{lower} = \frac{2}{k} \sum_{i=1}^{k-1} \sum_{j=i+1}^{k} (\Pr(C_i) + \Pr(C_j))\epsilon_{ij}.$$

### 4.3 Abductive Error Guarantees

Assume an algorithm selects from the set of possible causes $\mathcal{C}$ the cause with the highest estimated posterior. The preceding subsections detail the two possible sources of error:

1. Incomplete or imprecise background information (e.g., not knowing all the potential causes and causal relationships). This uncertainty is represented through $q$-percentile posterior confidence intervals in $\mathcal{U}$.

2. The true cause is not the cause with the highest true posterior. If the exact likelihood and prior is given, this minimum achievable error is simply expressed through the Bayes Error Rate (Definition 4.3).

We derive bounds of the error rate by combining these two possible sources of error. Let W denote the event of incorrect abduction (not selecting the true cause). Then, the probability of correctly selecting the maximum posterior $M_i$ and incorrect abduction is

$$\Pr(\text{W}, \text{IsMax}(M_i)) = \Pr(\text{W}|\text{IsMax}(M_i))\Pr(\text{IsMax}(M_i))$$
$$= \epsilon \Pr(\text{IsMax}(M_i)).$$

The probability of both incorrectly selecting the maximum posterior and incorrect abduction is

$$\Pr(\text{W}, \neg\text{IsMax}(M_i)) = \Pr(\text{W}|\neg\text{IsMax}(M_i))\Pr(\neg\text{IsMax}(M_i))$$
$$= (1 - \Pr(M_i|\mathbf{x}))(1 - \Pr(\text{IsMax}(M_i))).$$

Such definitions let us derive upper and lower bounds for the error rate assuming that all posteriors $M_i \in \mathcal{M}$ lie in $q$-percentile confidence intervals $[l_i, u_i] \in \mathcal{U}$ with probability 1. Let $\gamma_i$ denote the error rate given this assumption.

**Theorem 4.6.** *Let $\gamma_i$ denote the error rate of selected cause $C_i$ when assuming posterior $M_i$ lies in confidence interval $[l_i, u_i]$ almost surely. Then, $\gamma_i$ is bounded above by*

$$\gamma_i \leq \epsilon_{upper}\Pr(IsMax(M_i)) + (1 - l_i)(1 - \Pr(IsMax(M_i)))$$

*where $\epsilon_{upper}$ may be derived by Definition 4.4*

**Theorem 4.7.** *Let $\gamma_i$ denote the error rate of selected cause $C_i$ when assuming posterior $M_i$ lies in confidence interval $[l_i, u_i]$ almost surely. Then, $\gamma_i$ is bounded below by*

$$\gamma_i \geq \epsilon_{lower}\Pr(IsMax(M_i)) + (1 - u_i)(1 - \Pr(IsMax(M_i)))$$

*where $\epsilon_{lower}$ may be derived by Definition 4.5*

We extend this result to the general case where all posteriors $M_i \in \mathcal{M}$ are assumed to jointly lie in their respective confidence intervals $[l_i, u_i] \in \mathcal{U}$ with probability $q$.

**Theorem 4.8.** *Let $q^k$ be the probability that all $M_i \in \mathcal{M}$ lie in their respective confidence bounds $[l_i, u_i] \in \mathcal{U}$. Let $\gamma_{i,\ upper}$ be the upper bound of $\gamma_i$ defined in Theorem 4.6. Then, the upper bound of the general error rate is given by*

$$\Pr(W) \leq 1 - q^k(1 - \gamma_{i,\ upper}).$$

**Theorem 4.9.** *Let $q^k$ be the probability that all $M_i \in \mathcal{M}$ lie in their respective confidence bounds $[l_i, u_i] \in \mathcal{U}$. Let $\gamma_{i,\ lower}$ be the lower bound of $\gamma_i$ defined in Theorem 4.7. Then, the lower bound of the general error rate is given by*

$$\Pr(W) \geq \gamma_{i,\ lower} q^k.$$

We note that the upper bound $1 - q^k(1 - \gamma_{i,upper}) < 1$ and the lower bound $\gamma_{i,lower} q^k > 0$, so our bounds for $\Pr(W)$ are nontrivial, being strictly tighter than the general bounds on probabilities (e.g., $[0,1]$).

We should note that the bounds presented in this section assume noiseless observations. That is, we assume observation $\mathbf{x}$ is a wholly accurate description of the "true" outcomes of a cause. A noisy observation vector may have entries that deviate from the "true" outcome of a cause, akin to the possibility of a faulty observer or inaccurate data pipeline with which observations is processed (i.e., faulty equipment, random errors in sampling, etc.). Accounting for noisy observations for selecting the highest posterior cause is a subject of future work, and may involve the averaging of posteriors among a probability distribution of observation vectors.

The next section explores a different set of bounds describing the selection of *any* cause whose probability is above some threshold. With this broader definition of "success," we can account for noisy observations through applying the Algorithmic Search Framework (Montañez, 2017).

# 5 Search and Heuristic Applications

The Algorithmic Search Framework (ASF) characterizes learning problems as search, allowing one to equate the chance of success of any learning algorithm to that of a search process described by the three-tuple $(\Omega, T, F)$ – the *search space*, *target set*, and *external information resource*, respectively (Montañez, 2017). This framework formalizes the seminal work of Mitchell (1982) and extends results beyond binary classification problems (Montanez, 2017). Recent developments have also extended the ASF for continuous or fuzzy measures of success (Knell et al., 2024), allowing even greater flexibility. Most relevant to our use-case, the ASF provides formal bounds accounting for noise, and formalizes insights into the frequency of favorable search strategies and problems (Montanez, 2017). To our knowledge, there are no extensions of Mitchell's work or other formal frameworks that provide such use cases (Mitchell et al., 1986; Dupont et al., 1994; Duarte et al., 2023).

We have previously discussed abductive success in terms of finding the one "true" cause for some observation vector (which may or may not have the highest posterior) *assuming* the selection of the single highest posterior. Furthermore, we assumed noiseless observations. By reframing the ASF for abduction, we describe an algorithm's ability to identify the cause(s) with posteriors above some threshold in terms of information-theoretic properties within $(\Omega, T, F)$ and generalize to noisy observation vectors.

As explained in section 2, there exist no formal bounds on abductive success to our knowledge, and the ASF has not yet been applied to abduction. Existing work involving abduction and search such as abduction as inference to the best explanation (IBS) (Schurz, 2007), have not yielded formal explanations of abductive certainty or heuristics like Theorems 5.2 and 5.1. The ASF has not yet been applied to abduction, we believe doing so provides new formal and rigorous insights.

## 5.1 ASF: Success of Abduction through Search

We define each term of $(\Omega, T, F)$ as follows.

**Search Space ($\Omega$)** constitutes the finite set of pre-selected, plausible causes for the given observation vector $\mathbf{x}$; it is synonymous with $\mathcal{C}$ defined in 3.2. $P_i$ over search space $\Omega$ denotes the probability distribution over the space at step $i$, and $P_i(T)$ is the probability of success – namely, the amount of probability mass placed on the target set $T$ at time $i$ (Montañez, 2017). In our adaptation, $P_i$ denotes the posterior distribution of $\Pr(C_i|\mathbf{x})$ over all possible causes $C_i$ in $\Omega$. $P_i$ may be derived from aforementioned bounds $[l_i, u_i] \in \mathcal{U}$ of posterior-adjacent value $\Pr(C_i|\mathbf{x})\Pr(\mathbf{x})$ (Definition 4.1) with two modifications: (1) $P_i$ denotes the *point estimate*

*probability* of the posterior within these confidence bounds, and (2) this point estimate of $\Pr(C_i|\mathbf{x})\Pr(\mathbf{x})$ is inversely scaled by $\Pr(\mathbf{x})$ such that $P_i$ is a valid probability mass function that sums to one.

**Target Set ($T$)**, a subset of the search space $\Omega$, contains the set of the "more plausible" causes with posterior probability $P_i$ above or at *minimum performance value* in $(0,1]$. Search aims to identify causes in $\Omega$ that lie in $T$, a task whose difficulty increases as the threshold for $T$ rises. Note that $T$ is a random variable as it describes a set of likely causes for random observations.

**External Information Resource ($F$)** is a finite-length binary string drawn from a distribution with an "API"-like interface, meaning one can extract information from $F$ (Montañez, 2017). In our case, it embeds (1) the observation vector $\mathbf{x}$ whose cause we determine, and (2) the upper and lower bounds of the $q$-confidence intervals for likelihood and prior probabilities across $\Omega$ for every cause $C_i \in \Omega$. More specifically, $F$ contains the likelihood bounds $l_U(\mathbf{c},\mathbf{x})$ and $l_L(\mathbf{c},\mathbf{x})$ and prior bounds $r_U(\mathbf{c},\mathbf{x})$ and $r_L(\mathbf{c},\mathbf{x})$, which inform the construction posterior probability distribution $P_i$ over $\Omega$ for the search process as defined previously. Since $F$ is a function of random data, it is itself a random variable.

Note that, as explained in Section 4, the ASF places few restrictions on information resources $F$, and thus allows for both noisy or noiseless observations.

Framing abduction through the ASF, we apply established derivations of the maximal success probability of success defined in terms of information-theoretic properties of $(\Omega, T, F)$ and the complexity of the search problem (Montañez, 2017).

**Theorem 5.1.** *(Montañez, 2017) The probability of a successful abduction, $q$, is bounded above by*

$$q \le \frac{I(T;F) + D(P_T\|\mathcal{U}_T) + 1}{I_\Omega},$$

*where $I_\Omega = -\log\frac{|T|}{|\Omega|}$, $D(P_T\|\mathcal{U}_T)$ is the Kullback-Leibler divergence between the marginal distribution on target sets and the uniform distribution on possible target sets, and $I(T;F)$ is the mutual information between the target and observation.*

We interpret $I(T;F)$ as the dependence between the target set and the observation, $D(P_T\|\mathcal{U}_T)$ as the non-uniformness of the target, and $I_\Omega$ as the sparseness of the targets inside the search space. When the true cause is highly correlated with the observations (i.e., less random), the achievable success rate is high. When the search space consists of a large number of causes, the achievable success rate is lower. This gives us an additional information-theoretic upper bound on the probability of successful abduction.

## 5.2 ASF: High-Likelihood Causes are Rare

Any high-posterior cause must also confer high-likelihood to observed effects, due to the multiplicative nature of posterior computation. Yet a cause can only make an observation vector more probable at the cost of making others less probable. Such high-likelihood causes must necessarily be rare to the degree they confer high joint-probability on the observations, as shown by the following theorem (Montañez, 2017).

**Theorem 5.2.** *(Famine of Favorable Strategies Theorem, (Montañez, 2017)) For any fixed search problem $(\Omega, T, F)$, set of probability mass functions $\mathcal{P} = \{P : P \in [0,1]^{|\Omega|}, \sum_j P_j = 1\}$, and a fixed threshold $q_{\min} \in [0,1]$,*

$$\frac{\mu(\mathcal{G}_{t,q_{\min}})}{\mu(\mathcal{G}_\mathcal{P})} \le \frac{p}{q_{\min}},$$

*where $p = \frac{|T|}{|\Omega|}, \mathcal{G}_{t,q_{\min}} = \{P : P \in \mathcal{P}, t^\top P \ge q_{\min}\}$, and $\mu$ is Lebesgue measure.*

In contrast to Section 5.1, we consider a different search problem in applying Theorem 5.2. The search space $\Omega$ no longer consists of posteriors, but is now the space of all possible observation vectors, some of which are "close enough" to the true vector to comprise a noisy target set, $T$. Causes sample observation vectors by producing effects: a blind, weighted search. $F$ becomes irrelevant. Theorem 5.2 then tells us that the proportion of causes which confer at least $q_{\min}$ probability to the observation set is necessarily small whenever $q_{\min}$ is high, if we are only willing to tolerate so much noise in our observations (leading to small $|T|$).

One might argue that although not many causes can confer high *joint* likelihood to the observations, several independent causes might together constitute an abductive explanation for the observed phenomena, if each sufficiently raises the likelihood of a *single* observed feature. Simple arithmetic renders this possibility unpersuasive. Assuming independent causes for each observed feature, the probability of jointly occurring outcomes in an observation vector **x** scales exponentially with $|\mathbf{x}|$ or the number of features. For instance, if two features have a 50/50 chance of occurring coincidentally, then the chance of them occurring together is $1/2 \cdot 1/2 = 1/4$. For four such features, the probability drops to 6.25%. Thus, the coincidental co-occurrence of independent causes that together explain an observation vector is unlikely as the number of observations increases.

### 5.3 Increasing Certainty in Abductive Inference

Inductive inference error guarantees derive their strength from data abundance: increasing the number of observed examples typically increases the tightness of such bounds. In contrast, abductive inference proceeds from a single observation. How do we increase confidence in our abductive judgment? In the real world, our confidence in abductive reasoning typically depends on the amount of evidence supporting or contradicting a potential hypothesis. Though consisting of a single example, there are often many features of that observation, which may or may not be well-explained by a proposed cause. This suggests a "horizontal" mode of confirmation built on many conditionally independent features, rather than the "vertical" mode of confirmation based on many observed examples typical of inductive inference. We note the importance of conditional independence among features, since features that necessarily imply each other even given the cause do not give us additional confidence in our abductive judgment.

Recall that observation vector $\mathbf{x} \in \mathcal{O}$ consists of binary features representing the existence or non-existence of some conditionally independent observed outcome. Letting $x_1, \ldots, x_n$ represent each feature of $\mathbf{x} \in \mathcal{O}$ where $|\mathbf{x}| = \dim(O) = n$, we quantitatively demonstrate this phenomenon with the following theorem.

**Theorem 5.3.** *For each conditionally independent feature $x_1, \ldots, x_n$, define $\beta_i > 0$ such that for all $i = 1...n$,*

$$\Pr(x_i|C) = \beta_i \Pr(x_i|\overline{C}).$$

*Let $\beta = \sqrt[n]{\prod_i^n \beta_i}$, the geometric mean of the $\beta_i$. If $\beta > 1$, then*

$$\lim_{n \to \infty} \frac{\Pr(x_1, \ldots, x_n|C)}{\Pr(x_1, \ldots, x_n|\overline{C})} = \lim_{n \to \infty} \beta^n = \infty.$$

Each conditionally independent observation feature can either support ($\beta_i > 1$) or contradict ($\beta_i < 1$) the proposed cause. If features support the current cause $C$ on average (i.e., $\beta > 1$), then the confidence of abduction (ratio between likelihood under $C$ over $\overline{C}$) approaches infinity as the number of (on average) supporting features increases.

## 6 Discussion

We formalize abduction as selecting the cause with the highest estimated posterior from some finite pool of causes. Our focus on single-cause abduction problems is justified by their foundational role in simplifying complex decision-making processes, allowing for more precise modeling and analysis that lays the groundwork for tackling multi-cause scenarios with greater accuracy in future research. For $k$ possible causes whose posteriors are estimated within a confidence interval set with joint probability $q^k$, the probability of incorrect abduction $\Pr(W)$ is bounded below by $\Pr(W) \geq \gamma_{i,lower}q^k$ (Theorem 4.9) and bounded above by $\Pr(W) \leq 1 - q^k(1 - \gamma_{i, \text{upper}})$ (Theorem 4.8). As $q$ approaches 1, the bounds on the error rate depend more heavily on $\gamma_i$ (Theorems 4.6, 4.7), which scales with the Bayes Error Rate and the amount of overlap between uncertainty intervals. One should obtain comprehensive and representative training data (i.e., maximizing $q$) to achieve better estimates of posteriors and thus minimize error.

Extending this formalization to the ASF, we re-frame abductive success in information-theoretic terms and account for noisy observations. In this case, the maximum success rate of abduction is governed by the

complexity of the search problem and other information-theoretic properties (Theorem 5.1). The maximum success rate increases as the plausible causes (i.e., causes whose posteriors are above the *minimum performance value*) become more explainable and less random; inherent unpredictability is brought from the randomness of the "true" cause. However, one may constrain this randomness by decreasing the sparseness of the search space and/or excluding less probable causes.

Regarding the practicality of our results, it has been shown that bounds on the Bayes Error Rate can be empirically estimated by learning from training data instead of density estimation (Sekeh et al., 2020). Unlike traditional methods for estimating BER, such as those based on pairwise HP divergence or generalized Jensen-Shannon (JS) divergence, which becomes computationally infeasible as the number of classes or dimensions increases. The GHP-based method is shown to be computationally more efficient, making it more suitable for large-scale applications like neural networks (Sekeh et al., 2020). Then, in practice, it is possible to model a selective abduction problem using a Bayesian Neural Network and obtain approximate posterior distributions (Myshkov & Julier, 2016; Charnock et al., 2022), which can be directly used in our bounds for abductive inference.

The mathematical formalization and bounds established in our paper have implications for human-like reasoning abilities which are crucial for understanding the limits of decision-making processes in artificial intelligence. Theorem 5.2 demonstrates how high-likelihood causes are rare; one is less likely to stumble across them accidentally. In addition, more supporting observations increase our confidence in a unified causal explanation, instead of the coincidental co-occurrence of observed effects. Furthermore, Theorem 5.3 aims to capture the degree of certainty of our everyday abductive inferences. Consider a scenario where we are trying to convict a suspect of a crime. If pieces of evidence collectively support that the victim is guilty, our confidence to convict grows as the amount of such evidence grows. However, if pieces of evidence were heavily contradictory and/or refuted a suspect's involvement, then we become less confident of a conviction. Our confidence would approach 0 as the amount of (on average) contradictory observations tends toward infinity.

## 7 Conclusion

Abductive reasoning is a key component of human rationality and discovery. State-of-the-art artificial intelligence is currently incapable of performing abductive reasoning at a human level. To achieve true human-like reasoning, it is important to consider the process of abduction and incorporate such ability in future developments.

Our work formalizes selective abduction, deriving formal error guarantees for abductive reasoning within a finite space of causes. Also, by viewing selective abduction through the lens of the Algorithmic Search Framework, we better understood how the inherent complexities of abductive inference problems affect the achievable success rates. Future work might explore creative abduction using our framework as a starting point. Creative abduction can be represented through a search space that is potentially infinite. Rather than filtering $\mathcal{S}$ to a finite pool $\mathcal{C}$, we represent hypothesis generation as optimization within an *infinite* subset of $\mathcal{S}$. Proving bounds within this infinite set requires more complex mathematics, but extends the same underlying logic. Statistical bounds within such a framework would hold implications for general scientific reasoning and human creativity.

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

## Appendix: Proofs

**Proposition 4.1.** *Assuming that every $M_i \in \mathcal{M}$ lies in respective q-percentile confidence interval $[l_i, u_i] \in \mathcal{U}$, the max posterior is bounded by $u$ and $l$.*

*Proof.* We prove that the maximum posterior probability $M_{max} \in \mathcal{M}$ is in the interval $[l, u]$ (4.1) by way of contradiction. Assume $M_{max} \notin [l, u]$. Then, knowing that $|\mathcal{M}| \geq 2$, one of the following is true:

Case 1: $M_{max} > u$. Since $u$ is defined as the maximum of all posterior upper bounds, we reach a contradiction if $M_{max} > u$ as $M_{max}$ would not be in $\mathcal{M}$.

Case 2: $M_{max} < l$. If $M_{max} < l$, then since $l$ is defined as the highest lower bound, there must exist an $M_i \in \mathcal{M}$ such that $M_i \neq M_{max}$ whose lower bound $l_i \geq l$. If this is the case, $M_i > M_{max}$ since $M_i \geq l_i$, $l_i \geq l$, and $l > M_{max}$. $M_{max}$ would not be the maximum posterior, resulting in a contradiction.

Both possibilities result in contradiction, so $M_{max} \notin [l, u]$ is not true. Thus, the maximum posterior is bounded by $u$ and $l$, or $M_{max} \in [l, u]$. □

**Theorem 4.2.** *Let $\mathcal{M}' \subseteq \mathcal{M}$ denote the set of posteriors whose confidence intervals intersect with $[l, u]$. The probability that $M_i \in \mathcal{M}'$ is the maximum posterior is as follows:*

$$\Pr(IsMax(M_i)) = \int_l^u P\left(M_i = x, \bigcap_{\substack{M_j \in \mathcal{M}', \\ C_j \neq C_i}} (M_j < x)\right) dx.$$

*Proof.* Via Proposition 4.2.1, the max posterior is bounded by $u$ and $l$. So, if $x$ is the value of $M_i$ and $M_i$ is the maximum posterior, $l \leq x \leq u$. For $M_i$ to be the maximum posterior value with a value of $x$, both $M_i = x$ and $M_j < x$ for all $j \neq i$. So, we express $\Pr(\text{IsMax}(M_i))$ as an integral of joint probabilities:

$$\Pr(\text{IsMax}(M_i)) = \int_l^u \Pr\left(M_i = x, \bigcap_{j=1, j \neq i}^k (M_j < x)\right) dx.$$

□

**Theorem 4.6.** *Let $\gamma_i$ denote the error rate of selected cause $C_i$ when assuming posterior $M_i$ lies in confidence interval $[l_i, u_i]$ almost surely. Then, $\gamma_i$ is bounded above by*

$$\gamma_i \leq \epsilon_{upper} \Pr(IsMax(M_i)) + (1 - l_i)(1 - \Pr(IsMax(M_i)))$$

*where $\epsilon_{upper}$ may be derived by Definition 4.4*

*Proof.* Let $\Pr(W_i)$ be the probability we produce a wrong abduction given we selected cause $C_i$ (this is synonymous with the "error rate" of selecting cause $C_i$). By the law of total probability,

$$\Pr(W_i) = \Pr(W_i, \text{IsMax}(M_i)) + \Pr(W_i, \neg \text{IsMax}(M_i)).$$

We first discuss the case where the maximum posterior is selected (i.e., $\text{IsMax}(M_i)$ holds). Here, $\Pr(W_i \mid \text{IsMax}(M_i))$ is given by Bayes error rate, which is upper bounded by $\epsilon_{upper}$ (Definition 4.4). Thus,

$$\Pr(W_i, \text{IsMax}(M_i)) = \Pr(W_i | \text{IsMax}(M_i)) \Pr(\text{IsMax}(M_i))$$
$$\leq \epsilon_{upper} \Pr(\text{IsMax}(M_i)).$$

In the case that the highest posterior is *not* selected, the probability that we result in a false inference is given by $1 - \Pr(C_i | \mathbf{x})$, where $\Pr(C_i | \mathbf{x})$ is the theoretical true posterior. Let $l_i$ denote the lower bound of

$\Pr(C_i|\mathbf{x})$, which we assume to bound the true posterior almost surely. Thus,

$$\begin{aligned}
\Pr(W_i, \neg\text{IsMax}(M_i)) &= \Pr(W_i \mid \neg\text{IsMax}(M_i))\Pr(\neg\text{IsMax}(M_i)) \\
&= (1 - \Pr(C_i \mid \mathbf{x}))(1 - \Pr(\text{IsMax}(M_i))) \\
&\leq (1 - l_i)(1 - \Pr(\text{IsMax}(M_i))).
\end{aligned}$$

We combine our bounds to obtain

$$\Pr(W_i) \leq \epsilon_{upper}\Pr(\text{IsMax}(M_i)) + (1 - l_i)(1 - \Pr(\text{IsMax}(M_i))).$$

We relabel $\gamma_i$ as the error rate of selected cause $C_i$:

$$\gamma_i \leq \epsilon_{upper}\Pr(\text{IsMax}(M_i)) + (1 - l_i)(1 - \Pr(\text{IsMax}(M_i))).$$

$\square$

**Theorem 4.7.** *Let $\gamma_i$ denote the error rate of selected cause $C_i$ when assuming posterior $M_i$ lies in confidence interval $[l_i, u_i]$ almost surely. Then, $\gamma_i$ is bounded below by*

$$\gamma_i \geq \epsilon_{lower}\Pr(IsMax(M_i)) + (1 - u_i)(1 - \Pr(IsMax(M_i)))$$

*where $\epsilon_{lower}$ may be derived by Definition 4.5*

*Proof.* Let $\Pr(W_i)$ denote the probability we produce a wrong abduction given we selected cause $C_i$ (this is synonymous with the "error rate" of selecting cause $C_i$). By the law of total probability,

$$\Pr(W_i) = \Pr(W_i, \text{IsMax}(M_i)) + \Pr(W_i, \neg\text{IsMax}(M_i)).$$

We first explore the case where the highest posterior is selected (i.e., $\text{IsMax}(M_i)$ holds). Here, $\Pr(W_i \mid \text{IsMax}(M_i))$ is given by the Bayes error, which is lower-bounded $\epsilon_{lower}$ from Definition 4.5.

$$\begin{aligned}
\Pr(W_i, \text{IsMax}(M_i)) &= \Pr(W_i \mid \text{IsMax}(M_i))\Pr(\text{IsMax}(M_i)) \\
&\geq \epsilon_{lower}\Pr(\text{IsMax}(M_i))
\end{aligned}$$

If the cause we have selected does not have the maximum posterior (i.e., $\text{IsMax}(M_i)$ does not hold), the probability that we result in a false inference is given by $1 - \Pr(C_i|\mathbf{x})$, where $\Pr(C_i|\mathbf{x})$ is the theoretical true posterior. Let $u_i$ denote the upper bound of $\Pr(C_i|\mathbf{x})$ so that $1 - u_i$ lower bounds $1 - \Pr(C_i|\mathbf{x})$. Thus, we can derive the lower bound

$$\begin{aligned}
\Pr(W_i, \neg\text{IsMax}(M_i)) &= (1 - \Pr(C_i \mid \mathbf{x}))(1 - \Pr(\text{IsMax}(M_i))) \\
&\geq (1 - u_i)(1 - \Pr(\text{IsMax}(M_i))).
\end{aligned}$$

We combine the lower bounds of both components to obtain

$$\begin{aligned}
\Pr(W_i) &= \Pr(W_i, \text{IsMax}(M_i)) + \Pr(W_i, \neg\text{IsMax}(M_i)) \\
&\geq \epsilon_{lower}\Pr(\text{IsMax}(M_i))(1 - u_i)(1 - \Pr(\text{IsMax}(M_i))).
\end{aligned}$$

We then relabel $\gamma_i$ for error rate $\Pr(W_i)$:

$$\gamma_i \geq \epsilon_{lower}\Pr(\text{IsMax}(M_i)) + (1 - u_i)(1 - \Pr(\text{IsMax}(M_i))).$$

$\square$

**Theorem 4.8.** *Let $q^k$ be the probability that all $M_i \in \mathcal{M}$ lie in their respective confidence bounds $[l_i, u_i] \in \mathcal{U}$. Let $\gamma_{i,\ upper}$ be the upper bound of $\gamma_i$ defined in Theorem 4.6. Then, the upper bound of the general error rate is given by*

$$\Pr(W) \leq 1 - q^k(1 - \gamma_{i,\ upper}).$$

*Proof.* Let $CI_i$ be shorthand for the posterior confidence interval $[l_i, u_i] \in \mathcal{U}$ containing posterior $M_i \in \mathcal{M}$. By the law of total probability,

$$\Pr(W) = \Pr(W, \forall(M_i \in \mathcal{M}), M_i \in CI_i) + \Pr(W, \exists(M_i \in \mathcal{M})M_i \notin CI_i)$$
$$= \Pr(W \mid \forall(M_i \in \mathcal{M})M_i \in CI_i) \cdot \Pr(\forall(M_i \in \mathcal{M})M_i \in CI_i) +$$
$$\Pr(W \mid \exists(M_i \in \mathcal{M})M_i \notin CI_i) \cdot \Pr(\exists(M_i \in \mathcal{M})M_i \notin CI_i).$$

Note that the true posterior values are fixed and not random, but their estimates and confidence intervals (based on sampled data) *are* random. Given the true posterior values, data is generated from which confidence intervals are constructed and point estimates taken. The true posterior values thus act as parameters in a parameter estimation task. Given the value of such a parameter, the probability that a generated dataset and subsequent confidence interval captures the true parameter value is $q$, which (by d-separation) is conditionally independent of anything else that happens in the world. Specifically, any other parameter's (i.e., posterior's) value does not affect the probability that data generated using *this* parameter's value produces a confidence interval that captures it. All that matters is the specific parameter under which the data is generated. In other words, the probability that a second dataset generated from a different posterior produces a confidence interval that captures this second parameter's true value is **independent** of the outcome of the first data generation event, once we condition on the parameter. This second parameter is indeed given, as we need it to generate the data. Therefore, assuming $k$ confidence intervals are constructed from data conditioned on their true parameter values, the joint probability of all $k$ posterior probabilities being captured simultaneously by their respective $q$-percent confidence bounds is $q^k$. Thus,

$$\Pr(W) = \Pr(W \mid \forall(M_i \in \mathcal{M})M_i \in CI_i)q^k + \Pr(W \mid \exists(M_i \in \mathcal{M})M_i \notin CI_i)(1 - q^k).$$

We apply $\gamma_{i,upper}$ from Theorem 4.6, which is the probability of incorrect abduction assuming that all posterior probabilities fall into their respective confidence intervals. Thus, $\Pr(W \mid \forall(M_i \in \mathcal{M}), M_i \in CI_i)$ is bounded above by $\gamma_{i,\text{upper}}$. Additionally, we simply upper bound $\Pr(W \mid \forall(M_i \in \mathcal{M})M_i \notin CI_i)$ by one. Thus, we conclude

$$\Pr(W) \leq \gamma_{i,\text{upper}}q^k + (1)(1 - q^k),$$

or equivalently

$$\Pr(W) \leq 1 - q^k(1 - \gamma_{i,\text{upper}}).$$

$\square$

**Theorem 4.9.** *Let $q^k$ be the probability that all $M_i \in \mathcal{M}$ lie in their respective confidence bounds $[l_i, u_i] \in \mathcal{U}$. Let $\gamma_{i, lower}$ be the lower bound of $\gamma_i$ defined in Theorem 4.7. Then, the lower bound of the general error rate is given by*

$$\Pr(W) \geq \gamma_{i, lower}q^k.$$

*Proof.* Let $CI_i$ be shorthand for the posterior confidence interval $[l_i, u_i] \in \mathcal{U}$ containing posterior $M_i \in \mathcal{M}$. By the law of total probability,

$$\Pr(W) = \Pr(W, \forall(M_i \in \mathcal{M})M_i \in CI_i) + \Pr(W, \exists(M_i \in \mathcal{M})M_i \notin CI_i)$$
$$= \Pr(W \mid \forall(M_i \in \mathcal{M})M_i \in CI_i) \cdot \Pr(\forall(M_i \in \mathcal{M})M_i \in CI_i) +$$
$$\Pr(W \mid \exists(M_i \in \mathcal{M})M_i \notin CI_i) \cdot \Pr(\exists(M_i \in \mathcal{M})M_i \notin CI_i).$$

Recall that the number of considered causes is $|\mathcal{C}| = |\mathcal{M}| = k$. Assuming all $k$ confidence intervals simultaneously capture their respective posterior values with joint probability $q^k$ (see discussion in the proof for Theorem 4.8), we obtain

$$\Pr(W) = \Pr(W \mid \forall(M_i \in \mathcal{M})M_i \in CI_i)q^k + \Pr(W \mid \exists(M_i \in \mathcal{M})M_i \notin CI_i)(1 - q^k).$$

We apply $\gamma_{i,\text{lower}}$ from Theorem 4.7, which is the probability of incorrect abduction, assuming all posterior probabilities fall into their respective confidence intervals. Thus, $\Pr(W \mid \forall(M_i \in \mathcal{M})M_i \in CI_i)$ is bounded below by $\gamma_{i,\text{lower}}$. Additionally, we simply lower bound $\Pr(W \mid \exists(M_i \in \mathcal{M})M_i \notin CI_i)$ by zero. Thus, we conclude

$$\Pr(W) \geq \gamma_{i,\text{lower}}q^k + (0)(1 - q^k),$$

or equivalently

$$\Pr(W) \geq \gamma_{i,\text{lower}}q^k.$$

$\square$

**Theorem 5.3.** *For each conditionally independent feature $x_1, \ldots, x_n$, define $\beta_i > 0$ such that for all $i = 1...n$,*

$$\Pr(x_i|C) = \beta_i \Pr(x_i|\overline{C}).$$

*Let $\beta = \sqrt[n]{\prod_i^n \beta_i}$, the geometric mean of the $\beta_i$. If $\beta > 1$, then*

$$\lim_{n \to \infty} \frac{\Pr(x_1, \ldots, x_n|C)}{\Pr(x_1, \ldots, x_n|\overline{C})} = \lim_{n \to \infty} \beta^n = \infty.$$

*Proof.* If

$$\Pr(x_i|C) = \beta_i \Pr(x_i|\overline{C}),$$

then

$$\beta_i = \frac{\Pr(x_i|C)}{\Pr(x_i|\overline{C})}.$$

So, we can write

$$\prod_i^n \beta_i = \prod_i^n \frac{\Pr(x_i|C)}{\Pr(x_i|\overline{C})} = \frac{\prod_i^n \Pr(x_i|C)}{\prod_i^n \Pr(x_i|\overline{C})}.$$

Since the features are conditionally independent,

$$\prod_i^n \Pr(x_i|C) = \Pr(x_1, ..., x_n|C)$$

and

$$\prod_i^n \Pr(x_i|\bar{C}) = \Pr(x_1, ..., x_n|\bar{C}).$$

Thus,

$$\beta^n = \prod_i^n \beta_i = \frac{\Pr(x_1, ..., x_n|C)}{\Pr(x_1, ..., x_n|\bar{C})}.$$

Therefore, when $\beta > 1$,

$$\lim_{n \to \infty} \frac{\Pr(x_1, ..., x_n|C)}{\Pr(x_1, ..., x_n|\bar{C})} = \lim_{n \to \infty} \beta^n = \infty.$$

$\square$

## Appendix: Toy Example and Additional Derivations of Theorem 4.2

### Setup

We present a toy example computing the bounds in section 4 (namely, Theorems 4.8 and 4.9). In this example, $\dim(\mathcal{O}) = 2$ and has features ["pill taken?", "headache relief?"]. We consider three selected possible causes: ["aspirin", "caffeine", "placebo"] and derive abductive error bounds for abducing the observation [1,0].

For simplicity, this example uses normalized posterior probabilities (i.e., $M_i = \frac{\Pr(\mathbf{x}|C_i)\Pr(C_i)}{\Pr(\mathbf{x})}$ rather than $\Pr(\mathbf{x}|C_i)\Pr(C_i)$). This produces the same results as using non-normalized posteriors since the ranking of posterior ranges is the same.

The setup of the example is as follows. Table 9 displays the evidence distribution over $\mathcal{O}$ (Definition 3.1)–the only given distribution assuming exact probabilities.

Table 3: Example evidence distribution.

| Pill taken? | Headache relieved? | $\mathbf{x}$ | $\Pr(\mathbf{x})$ |
|:---:|:---:|:---:|:---:|
| no | no | $\langle 0, 0 \rangle$ | 0.3 |
| no | yes | $\langle 0, 1 \rangle$ | 0.05 |
| yes | no | $\langle 1, 0 \rangle$ | 0.15 |
| yes | yes | $\langle 1, 1 \rangle$ | 0.5 |

Tables 4 and 5 display the $q = 0.95$ percentile prior and likelihood ranges for each possible cause: "aspirin", "placebo", and "caffeine".

Table 4: 95% confidence intervals for the prior of each cause.

| Cause | $\Pr(C_i)$ Lower Bound | $\Pr(C_i)$ Upper Bound |
|:---:|:---:|:---:|
| aspirin | 0.4 | 0.517 |
| caffeine | 0.32 | 0.37 |
| placebo | 0.1 | 0.113 |

Table 5: 95% confidence intervals for the likelihood of each cause.

| $\mathbf{x}$ | CI for $\Pr(\mathbf{x}|do(\text{aspirin} = \texttt{True}))$ | CI for $\Pr(\mathbf{x}|do(\text{placebo} = \texttt{True}))$ | CI for $\Pr(\mathbf{x}|do(\text{caffeine} = \texttt{True}))$ |
|:---:|:---:|:---:|:---:|
| $\langle 0, 0 \rangle$ | [0.02, 0.058] | [0.0067, 0.067] | [0.64, 0.83] |
| $\langle 0, 1 \rangle$ | [0.006, 0.014] | [0.004, 0.06] | [0.58, 0.8] |
| $\langle 1, 0 \rangle$ | [0.13, 0.15] | [0.33, 0.4] | [0.011, 0.067] |
| $\langle 1, 1 \rangle$ | [0.4, 0.77] | [0.005, 0.06] | [0.064, 0.14] |

### Calculating $\Pr(\textbf{IsMax}(M_i))$

From here, we will abduce the cause of observation $\mathbf{x} = [1, 0]$ ("pill taken", "headache not relieved"). The posterior confidence bounds for this observation are displayed in **??**. Note that for this toy example, we assume uniform distributions within posterior confidence intervals for easy visualization, but our methods and code support any bounded distribution.

Recall that $\Pr(\text{IsMax}(M_i))$ is the probability that $M_i$ is the maximum posterior *assuming* that all other posteriors $M_i \in \mathcal{M}$ lie in their $q$-percentile confidence intervals $[l_i, u_i]$. Theorem 4.2 presents a general method of describing this value, making no assumptions of the positions of the upper/lower posterior bounds. We derive two directly computable versions of Theorem 4.2 with varying constraints.

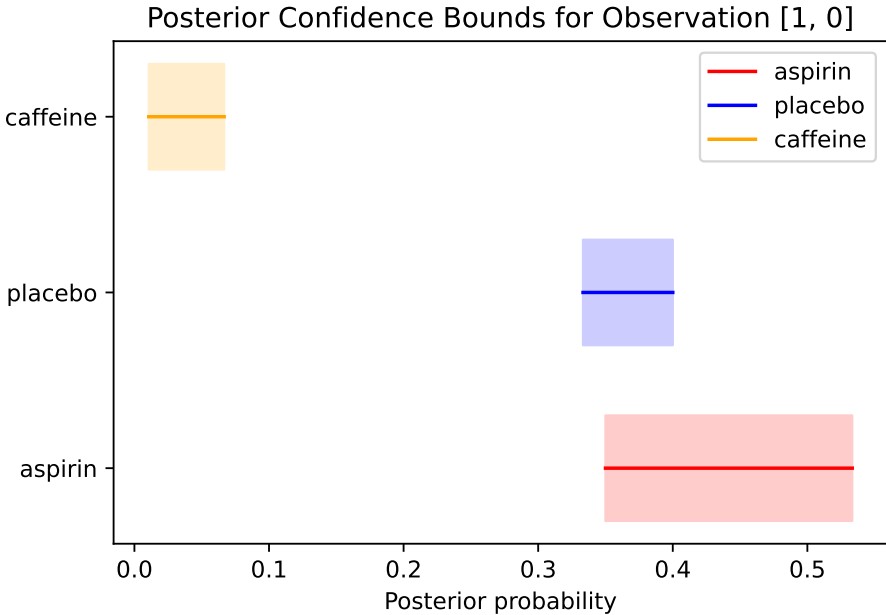

Figure 2: Posterior probability ranges for observation [1,0] (pill taken, headache not relieved).

If the sum of the upper bounds for each posterior does not exceed 1 (or, for non-normalized posteriors, $\Pr(\mathbf{x})$), the posteriors of selected causes are independent[6], leaving any remaining probability to the (dependent) last cause, $C_{other}$ with posterior $M_{other}$. This is the combined (likely small) posterior probability that any cause not in the selected set $\mathcal{C}$ is the true cause of $\mathbf{x}$. Representing each $M_i \in \mathcal{M}$ as a random variable (rv), then $M_{other} = 1 - \sum_{M_i \neq M_{other}} M_i$ where all $M_i \neq M_{other}$ are independent of each other. From here, we will denote any $M_1 \dots M_N$ as the posteriors not describing $C_{other}$ for notational simplicity. Let $f$ denote the joint probability density function of all $M_1 \dots M_N$, and let $f_j(m_j)$ denote the (give) marginal distribution of rv $M_j$.

One brute-force approach of computing theorem 4.2 is to treat $\Pr(\text{IsMax}(M_i))$ as the expected value of the following indicator function. Let $m_j$ denote the specific value taken by any posterior $M_j$ when computing the integral. Then, let the the following indicator function denote whether $m_i$ is the maximum posterior among all other posteriors' set values *and* the probability of $m_{other} = 1 = \sum_j m_j$.

$$\mathbf{1}_{m_i}(m_1, \dots, m_N) = \begin{cases} 1 & \text{if } m_i = \max\{m_1, \dots m_N, 1 - \sum_j m_j\} \\ 0 & \text{otherwise} \end{cases}$$

Intuitively, the expected value of this indicator function is similar to tallying up the total number of configurations of $m_1, \dots m_N$, where $m_i$ is the maximum, and then dividing it by the total number of configurations of $m_1, \dots, m_N$. This is equivalent to the probability that random variable $M_i$ is the maximum.

---

[6]If the sum of normalized posterior upper bounds exceeds 1, then posteriors cannot be treated as independent random variables because the configurations where their sum exceeds 1 are invalid with probability 0. A way to compute $\Pr(\text{IsMax}(M_i))$ when posterior upper bounds exceed 1 is through random sampling over the given distributions of $M_i \in \mathcal{M}$, ignoring instances where the sum of posteriors exceeds 1.

$$\Pr(\text{IsMax}(M_i)) = \int_{l_1}^{u_1} \cdots \int_{l_N}^{u_N} \mathbf{1}_{m_i}(m_1, \ldots, m_N) f(m_1, \ldots, m_N) \, dm_N \cdots dm_1$$

$$= \int_{l_1}^{u_1} \cdots \int_{l_N}^{u_N} \mathbf{1}_{m_i}(m_1, \ldots, m_N) \prod_{k=1}^{N} f(m_k) \, dm_N \cdots dm_1$$

Furthermore, if we can guarantee that $M_{other}$ cannot be the highest posterior when $M_i \in \mathcal{M}$ lie their $q$-percent bounds (i.e., there exists at least one posterior lower bound $l_i$ such that $l_i > 1 - \sum_j l_j$), then we can apply a more efficient computable derivation of Theorem 4.2. Note that $M_{other}$ would likely have this property for any problem with a non-trivial number of causes.

$$\Pr(\text{IsMax}(M_i)) = \int_{l}^{u} P\left(M_i = x, \bigcap_{j \neq i}^{N} (M_j < x)\right) dx \quad \text{(Theorem 4.2 with adjusted notation)}$$

$$= \int_{l}^{u} P(M_i = x) P\left(\bigcap_{j \neq i}^{N} M_j \leq M_i \,\Big|\, M_i = x\right) dx \quad \text{(``and" rule)}$$

$$= \int_{l}^{u} P(M_i = x) P\left(\bigcap_{j \neq i}^{N} M_j \leq x \,\Big|\, M_i = x\right) dx$$

$$= \int_{l}^{u} P(M_i = x) \prod_{j \neq i}^{N} P\left(M_j \leq x \,\Big|\, M_i = x\right) dx \quad \text{(Independence of posteriors } M_1 \ldots M_N)$$

If we have $l_j > x$ for some $j \neq i$, then and $M_j$ is guaranteed to be greater than $x$, and so $M_i$ cannot be the maximum posterior at $M_i = x$. Otherwise, we can find the probability that $M_j < x$ by integrating over the probability density function of $M_j$, $f_j$. This leaves the following computable form.

$$\Pr(\text{IsMax}(M_i)) = \int_{l}^{u} P(M_i = x) \prod_{j \neq i}^{N} \left[\mathbf{1}_{l_j \leq x} \int_{l_i}^{x} f_j(y) dy\right] dx$$

We used the derivation above to compute the following $\Pr(\text{IsMax}(M_i))$ values for each selected cause.

Table 6: $\Pr(\text{IsMax}(M_i))$ for observation [1,0].

| Cause | $\Pr(\text{IsMax}(M_i))$ | Error |
|-------|------------------------|-------|
| aspirin | 0.89772 | 1.0e-05 |
| caffeine | 0.10227 | 7.8e-10 |
| placebo | 0.0 | 0.0 |

Referring to the posterior ranges in Figure **??**, notice that "aspirin", as the rightmost posterior range, is the most likely to be the maximum posterior when all posteriors are within their confidence intervals. This is consistent with the calculation above, as aspirin has the highest probability of being the maximum posterior. Also, notice from Figure **??** that "caffeine" can never the greatest posterior as its upper bound is much lower than the lower bound of any other cause. As expected, this $\Pr(\text{IsMax}(\text{caffeine}))$ is calculated to be zero with no error.

**Bayes Error Rate, $\gamma_i$, and Final Bounds**

Next, we find Bayes Error Rate (BER). This example is small enough to directly calculate lower and upper bounds of BER with the summation form of Definition 4.3 (Sekeh et. al.) without the use of estimation

techniques. (For more complex problems with many causes $|\mathcal{O}| = 2^{|\mathcal{C}|}$ may be very large, so it is likely more practical apply the estimation techniques while treating the observations as continuous). We found Bayes' rate to be bounded by $\epsilon_{lower} = 0.23$ and $\epsilon_{upper} = 0.5667$ for the observation [1,0].

Now that we have $\epsilon_{lower}$, $\epsilon_{upper}$, and the $\Pr(\text{IsMax}(M_i))$ values for all possible causes, we can follow section 4.3 to find the final general error rate bounds.

We can now calculate $\gamma_{i,upper}$ and $\gamma_{i,lower}$ per Theorems 4.7 and 4.6 for each cause – the error rate when posterior $M_i$ is chosen, given the assumption that posteriors lie in their 95% confidence intervals.

$$\gamma_{i,upper} = \epsilon_{\text{upper}} \Pr(\text{IsMax}(M_i)) + (1 - l_i)(1 - \Pr(\text{IsMax}(M_i))),$$

$$\gamma_{i,lower} = \epsilon_{\text{lower}} \Pr(\text{IsMax}(M_i)) + (1 - u_i)(1 - \Pr(\text{IsMax}(M_i)))$$

Next, we calculate upper and lower bounds for this error rate for all three possible causes:

Table 7: $\gamma_i$ bounds for observation [1,0].

| Posterior | $\gamma_{i,lower}$ | $\gamma_{i,lower}$ |
|---|---|---|
| placebo | 0.562 | 0.656 |
| aspirin | 0.254 | 0.575 |
| caffeine | 0.933 | 0.989 |

As expected, aspirin (as the highest posterior range in Figure **??**) has the lowest error rate assuming posteriors lie in confidence intervals. Caffeine (as the lowest posterior range) has the highest error rate – i.e., choosing caffeine as the cause for observation [1,0] ("pill taken" and "no headache relief") is most likely to be wrong.

We now compute the final bounds per Theorems 4.9 and 4.8. Where $W$ is the event of "wrong abduction", $\Pr(W) \le 1 - q^N(1 - \gamma_{i,\text{upper}})$ and $\Pr(W) \ge \gamma_{i,\text{lower}}q^N$.

Table 8: General abductive error bounds for observation [1,0].

| Cause | $\Pr(W)$ lower bound | $\Pr(W)$ upper bound |
|---|---|---|
| placebo | 0.562 | 0.656 |
| aspirin | 0.254 | 0.575 |
| caffeine | 0.933 | 0.989 |

As expected, "aspirin" has the lowest error rate rate, followed by "placebo" and then "caffeine".

**Results Summary**

Table 9 displays the computed bounds for $\gamma_i$ and $\Pr(W)$ for every cause and observation. Figure 3 displays the corresponding posterior confidence interval ranges for each observation.

Table 9: Summary of bounds for every observation

| x | $\gamma_i$ (aspirin) | $\gamma_i$ (placebo) | $\gamma_i$ (caffeine) | $\Pr(W)$ (aspirin) | $\Pr(W)$ (placebo) | $\Pr(W)$ (caffeine) |
|---|---|---|---|---|---|---|
| $\langle 0,0 \rangle$ | [0.9, 0.973] | [0.93,0.99] | [0.23,0.57] | [0.772,0.977] | [0.8,0.994] | [0.197, 0.628] |
| $\langle 0,1 \rangle$ | [0.86,0.952] | [0.94,0.996] | [0.23, 0.567] | [0.737,0.959] | [0.806,0.997] | [0.197, 0.628] |
| $\langle 1,0 \rangle$ | [0.254,0.575] | [0.562,0.656] | [0.933,0.989] | [0.218, 0.8] | [0.482, 0.705] | [0.80,0.99] |
| $\langle 1,1 \rangle$ | [0.223, 0.567] | [0.94,0.995] | [0.86,0.936] | [0.197,0.629] | [0.806,0.996] | [0.737,0.945] |

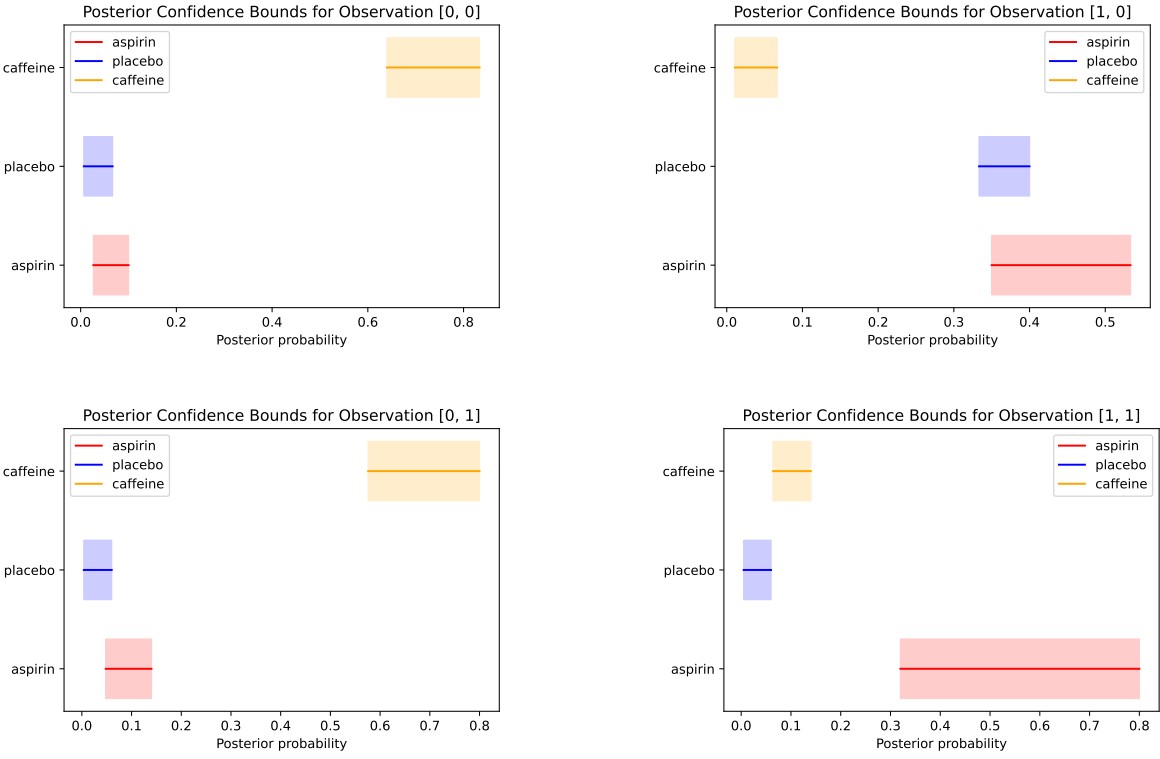

Figure 3: Posterior confidence ranges for each observation.

