# OpenReview forum: "Probabilistic Guarantees for Abductive Inference"
_TMLR — Rejected by TMLR_

### Review · Reviewer_HXov · 2024-06-12

**Summary Of Contributions:**

The article considers a sort of probabilistic abduction. It proposes an approach combining Bayesian probability and the Algorithmic Search Framework and defining different bounds for the probability of correct abductions.

**Audience:**

Yes

**Broader Impact Concerns:**

I do not see any ethical implication.

**Claims And Evidence:**

No

**Requested Changes:**

About weaknesses (all important regarding the acceptance of the article):
1. Novelty. Most of the results described in Section 5 are based on what is presented by (Montanez, 2017), but in my opinion the paper does not clearly describe which results it adds to these.

2. Related work section. In Section 2.3, the paper mentions Abductive logic Programming and Probabilistic Horn Abduction but I think some work on Probabilistic Abductive logic Programming is missing. For example, I can list:
T. Calin-Rares, M. Nataly, R. Alessandra, B. Krysia, On minimality and integrity constraints in probabilistic abduction, in: Logic for Programming, Artificial Intelligence, and Reasoning, Springer, 2013, pp. 759–775.
or the more recent work
Damiano Azzolini, Elena Bellodi, Stefano Ferilli, Fabrizio Riguzzi, and Riccardo Zese. Abduction with probabilistic logic programming under the distribution semantics. International Journal of Approximate Reasoning, 142:41--63, 2022.


3. Clarity of presentation. There are some parts that in my opinion should be better explained. For example, Section 3.2 does not clearly define the size of $\mathbf{x}$ and whether the size depends on $\mathcal{O}$. Furthermore, it is not clear how the probability mass is distributed and thus how the probability distribution is defined. For example, the paper should better clarify whether the probability distribution presented in Table 2 depends only on $\mathbf{x}$ and $C_i$ or whether it is fixed in the set of observations.
Furthermore, I would suggest introducing a running example (it could be the one in Section 3.2) on which to calculate the different bounds. This would improve the clarity of the presentation of the concepts.

4. Missing references. There are several references to concepts or definitions/theorems not present in the paper, which are probably a remaining from a previous, possibly extended, version of the submitted paper. For example, on page 9 the paper mentions Definitions 4.2.2 and 4.2.3 which are actually 4.4 and 4.5. The same problem occurs on page 10 for Definitions 3.4.1 and 4.1.1, which do not exist in the paper. Or in Section 6, where reference is made to small neural networks that are not mentioned elsewhere in the paper. For this last point in particular, it would be useful to discuss the meaning of this reference further.

**Strengths And Weaknesses:**

Strengths
1. The presented approach, if better described, could be interesting and give a good contribution to the abduction field.
2. Presence of proofs in the appendix.

Weaknesses
1. The novelty of the paper is not always well framed and described.
2. The Related Works Section is a bit incomplete.
3. The article does not always explain concepts clearly.
4. The paper should be carefully checked to correct missing references.

---

> ### Author Response · Authors · 2024-07-14
> **Response to Reviewer HXov**
>
> Thank you for your detailed feedback. We have attempted to address the specific raised issues as follows:
>
> >Novelty. Most of the results described in Section 5 are based on what is presented by (Montanez, 2017), but in my opinion the paper does not clearly describe which results it adds to these.
>
> 1. **Novelty:** Thanks for pointing this out. Our original submission did not clearly justify our choice in using the ASF and the novelty of the results produced. In the revised start to Section 5, we discuss how (1) no one has applied the ASF to the case of abduction, (2) existing adjacent approaches have not produced any insights into bounds for abductive success or heuristic justifications. We also clearly describe the additional contributions the ASF brings at the beginning of Section 5 for clarity.
>
> >Related work section. In Section 2.3, the paper mentions Abductive logic Programming and Probabilistic Horn Abduction but I think some work on Probabilistic Abductive logic Programming is missing. For example, I can list: T. Calin-Rares, M. Nataly, R. Alessandra, B. Krysia, On minimality and integrity constraints in probabilistic abduction, in: Logic for Programming, Artificial Intelligence, and Reasoning, Springer, 2013, pp. 759–775. or the more recent work Damiano Azzolini, Elena Bellodi, Stefano Ferilli, Fabrizio Riguzzi, and Riccardo Zese. Abduction with probabilistic logic programming under the distribution semantics. International Journal of Approximate Reasoning, 142:41--63, 2022.
>
>
> 2. **Related work:** Our revised draft goes over additional sources exploring probabilistic abductive logic programming and briefly summarizes how our work departs from it. To our knowledge, developments in this field (such as the work you listed) assume exact underlying distributions rather than confidence intervals, and do not provide bounds for abductive success. We need to be mindful of the two other reviewers who suggested that the first few sections needed to be cut down, but we will be adding more detailed explanations if space allows. (This subpoint has been edited recently since we misread your comment in our original response (apologies) -- thank you for pointing out additional work we could cover.)
>
> >Clarity of presentation. There are some parts that in my opinion should be better explained. For example, Section 3.2 does not clearly define the size of x and whether the size depends on O. Furthermore, it is not clear how the probability mass is distributed and thus how the probability distribution is defined. For example, the paper should better clarify whether the probability distribution presented in Table 2 depends only on x and C_i or whether it is fixed in the set of observations. Furthermore, I would suggest introducing a running example (it could be the one in Section 3.2) on which to calculate the different bounds. This would improve the clarity of the presentation of the concepts.
>
> 3. **Clarity:** To clarify your first point regarding the size of observation vector $\textbf{x}$, since it is a vector in space $\mathcal{O}$, the length/number of features in $\textbf{x}$ matches the dimension of $\mathcal{O}$. Would you recommend we include this explicitly? To the second point, in our revised draft, we provided an additional explanation in Section 3.2 describing that the likelihood probability mass function is specific only to the cause $C_i$ and then acts upon the observation space $\mathcal{O}$. Unfortunately, we could not provide extensive explanations due to the need to condense these earlier sections from other reviewers' feedback, but let us know if what we have seems sufficient. Regarding the recommendation about including a running example for bound calculations, we are currently working on extending the example from Section 3.2 to have example calculations for at least the first set of bounds (Section 4). This will be included in the appendix of our next revision due to space constraints.
>
> > Missing references. There are several references to concepts or definitions/theorems not present in the paper.
>
> 4. **Missing references:** Thank you for noticing these careless mistakes. We believe we have fixed all incorrect references in our revised draft. We have also provided proper references and an additional explanation about the estimation techniques with neural networks, formerly referred to as “small neural networks” (see paragraph 3 of section 6). In short, a Bayesian neural network can be used to model a selective abduction problem, which can then approximate posterior distributions.
>
> Please let us know your thoughts on the current revisions and what further changes are necessary. Thank you again for your attention to detail.

---

> > ### Comment · Reviewer_HXov · 2024-07-16
> > **Response to authors**
> >
> > Thank you for the response.
> >
> > *Point 2.* Regarding the related work, I think you get the point. In the two papers, the authors want to compute the most preferred possibly minimal solutions, returning exact probability values instead of intervals.
> >
> > *Point 3.* From the paper I understand the following. Assume that $\mathcal{O}$ has size $N$ with $N > 2$, if I concentrate on $C_i$ aspirin, the observations I have to consider are {“Pill taken?”, “Headache relieved?”}, but x must also consider the other observation vectors in $\mathcal{O}$ since x $\in\mathcal{O}$. In this case, the probability distribution must be defined over observations that are not of interest for $C_i$. Finally, I have to define a different distribution for each $C_i$ on every x in this setting. Am I right? If so, how should I define these distributions?
> > Thank you for the clarification.

---

> > > ### Author Response · Authors · 2024-07-16
> > > **Response to Review HXov**
> > >
> > > Thank you for your quick response!
> > >
> > > *Point 2.* Yes, this is what we found -- thank you for giving us additional sources. What we have written in our current draft is that (similar to Probabilistic Horn Abduction), these developments work with exact probability values rather than (more flexible) confidence intervals and do not provide the guarantees for success.
> > >
> > > *Point 3.* We think you are understanding the distribution setup correctly. For an example, assume that $\mathcal{O}$ has two dimensions representing {“Pill taken?”, “Headache relieved?”}, then $\mathcal{O}$ consists of all possible $2^2$ binary configurations of answering "yes" or "no" to the questions  {“Pill taken?”, “Headache relieved?”} such as the example for table 2. Every possible cause $C_i \in \mathcal{C}$ defines some likelihood probability distribution over the observation space and considers all possible observation vectors, concentrating mass around observations that cause is more associated with. For example, for the likelihood of $C_i$ = "aspirin", mass would be more concentrated around vector $[1,1]$, as aspirin is taken in pill form and usually relieves headaches. For $C_i =$ "placebo pill", mass would be more concentrated around vector $[1, 0]$. Thus, you have to define a different likelihood distribution for each $C_i$ acting on every $\mathbf{x} \in \mathcal{O}$.
> > >
> > > Thank you again for your feedback, please let us know if you recommend more revisions or if anything is unclear.

---

### Review · Reviewer_bTe3 · 2024-06-24

**Summary Of Contributions:**

The goal of this paper is to "establish two novel sets of
probabilistic bounds on the success of abduction when (1) selecting
the single most likely cause while assuming noiseless observations,
and (2) selecting any cause above some probability threshold while
accounting for noisy observations." Noisy observations are dealt with
using percentile-based confidence intervals.

**Audience:**

No

**Claims And Evidence:**

Yes

**Requested Changes:**

The issues mentioned under "SPECIFIC PROBLEMS" need addressing.

However, even if they were all fixed I think it improbable that a revised version (as opposed to something amounting to a new paper) would be acceptable.

**Strengths And Weaknesses:**

The writing is generally good and clear. I'm OK with a paper devoted
to abduction appearing in a machine learning journal. However, the
technical and concceptual contribution is far from being sufficiently significant
for publication in TMLR. In addition there are a number of specific
problems with the paper.

The paper could be condensed. There is a lot of text used to define /
analyse quite elementary things. Section 3.1 could be replaced by a
sentence which states that outcomes are binary vectors. Much of
Section 3.3 is elementary probability theory.

SPECIFIC PROBLEMS

It is stated that maximum likelihood estimation (MLE) "optimize[s]
model parameters by maximizing the posterior probability". But MLE
very much does not maximise posterior probability - it is a
non-Bayesian method with non-Bayesian justification. Also to say that
both MLE and MAP estimation "apply abductive reasoning" is to
generalise the meaning of "abductive reasoning" too much.

In any discussion of (probabilistic) causes one should at least
mention the distinction between P(x|aspirin=True) and
P(x|do(aspirin=True)) and related work on the do-calculus but this is
not done here. Doing so would have improved Section 3.2.

The ASF "characterizes learning problems as search" which is fine. But
there is a lot of work analysing this characterisation dating back to
"Mitchell, T. M. (1982). Generalization as search. Artificial
Intelligence, 18(2), 203–226." Perhaps (Montañez, 2017) relates ASF to
this body of work, but it would also be good here to state that
characterising learning as search is nothing new and what is
particularly useful about the ASF approach.

Section 5.1 : We are promised a definition of F but we don't get one,
merely that it "embeds" x and confidence intervals. Since we have
I(T;F) in Theorem 5.1 we might assume that T and F are both random
variables even though F somehow contains confidence intervals and T is
explicitly stated to be a subset (of the search space). No proof of
Theorem 5.1 is supplied, so we the reader cannot investigate F further
(presumably the proof is in Montañez, 2017).


SMALL POINTS

In Table 1, I think y^m and x^m need swapping since x^m is a
(possible) "cause" of observing y^m.

---

> ### Author Response · Authors · 2024-07-14
> **Response to Reviewer bTe3**
>
> Thank you for your feedback and detailed descriptions of specific issues. We have attempted to address the identified problems in our revised draft.
>
> > The paper could be condensed
>
> 1. **Paper length:** In our revised version, we have condensed much of Sections 3.1, 3.2, 3.3, and some of the related work.
>
> > MLE very much does not maximise posterior probability...to say that both MLE and MAP estimation "apply abductive reasoning" is to generalise the meaning of "abductive reasoning" too much.
>
>
>
> 2. **MLE Typo/Abduction oversimplication:**  Thank you for catching this error. An earlier draft went into more detail analogizing MLE to an abductive process, but it was cut for space leaving that typo. We have removed MLE entirely in our revised version. We also explain in a bit more detail how we are analogizing MAP to abduction, as MAP chooses which parameter setting (“cause”) maximizes the posterior with the data (“observations”). The purpose of including MAP was to emphasize how common abductive processes are in everyday data science techniques. This was intended to stress the possible applications for the presented bounds and formalizations.
>
> > In any discussion of (probabilistic) causes one should at least mention the distinction between P(x|aspirin=True) and P(x|do(aspirin=True)) and related work on the do-calculus
>
> 3. **Do-calculus:** We have added a quick sentence describing the distinction for do-calculus in 3.2 and apply the formalism in the example for Table 2.
>
> >  it would also be good here to state that characterising learning as search is nothing new and what is particularly useful about the ASF approach.
>
> 4. **Justifying the ASF:** Our submitted draft failed to clearly justify our choice to use the ASF -- thank you for pointing this out. We have provided such justifications at the start of Section 5 of our revised version. In short, (Montanez, 2017) formalizes work by Mitchell and extends applications beyond binary classification problems. Unlike work by Mitchell and other frameworks (to our knowledge), the ASF provides bounds that account for noise (Theorem 5.1), in addition to formal insights into the frequency of favorable strategies (or, in our case, high-likelihood causes) for Theorem 5.2.
>
> > Section 5.1 : We are promised a definition of F but we don't get one
>
> 5. **Definition of $F$:** Our revised draft adds more explanation of what $F$ is in Section 5.1, namely a finite length binary string drawn from a distribution, with an "API"-like interface from which you can extract information. The proof is in (Montanez, 2017). Our initial submission presented this in a confusing way, so we appreciate this point.
>
> > In Table 1, I think y^m and x^m need swapping since x^m is a (possible) "cause" of observing y^m.
>
> 6. **Table 1:**  Assuming this is referring $x^m$ and $y_m$ in the “classification” section (please correct us if this is wrong), $y_m$ represents the correct class/cause and $x^m$ represents the input data/observation vector. We are describing the application of a trained model acting on an input vector $x^m$ to determine the correct class ($y_m$). Is this the issue you were referring to? If so, please let us know if there is an issue with our logic or if we should give more explanation.
>
> > However, the technical and conceptual contribution is far from being sufficiently significant for publication in TMLR.
>
> 7. **Contribution significance:** As discussed in section 2, there is no existing work deriving error bounds for successful abduction. Given the ubiquity of abductive processes in data science, and its larger significance in the scientific process and intelligence, we initially believed that producing novel error guarantees and a formal framework for abduction were worthwhile contributions that at least some of TMLR's audience may be interested in, per [TMLR's acceptance criteria](https://jmlr.org/tmlr/acceptance-criteria.html):
>  > Crucially, it should not be used as a reason to reject work that isn't considered “significant” or “impactful”...We explicitly avoid these terms (“significant”, “impactful”, “novel”), and focus instead on the notion of “interest”. If the authors make it clear that there is something to be learned by some researchers in their area from their work, then the criterion of interest is considered satisfied.
>
>     **Could you please provide more specifics about how our current technical and conceptual contributions could be improved to meet TMLR's acceptance criteria for audience interest?**
>
>     Regarding recent edits related to novelty/contribution, we have added discussion of novelty to the beginning of section 5 in response to reviewer HXov. We also corrected unclear writing in section 3.4, and our new submission clarifies that our framework can account for multiple concurrent causes.
>
> Thank you again for the feedback, please let us know your thoughts.

---

### Review · Reviewer_HdgV · 2024-07-06

**Summary Of Contributions:**

The paper combined Bayesian decision theory and algorithmic search framework for a quantitative formalization of abductive reasoning. Two settings are considered - selecting the single mostly likely abductive cause while assuming no noise in the observations, and selecting causes above a threshold of probability when the observations are noisy. The proposed approach only assumes percentle-based CIs for the underlying prior and likelihood distributions.

**Audience:**

No

**Broader Impact Concerns:**

There are no broader impact concerns.

**Claims And Evidence:**

Yes

**Requested Changes:**

* Much of the introductory material in the first 5 pages can be shortened.

* Could you please add discussion to address the weaknesses mentioned above?

**Strengths And Weaknesses:**

Strengths:

- Taking into account the noise in observations makes the abductive inference formulation more practical.

-  To account for uncertainty, the paper estimates likelihood, prior, and posterior probabilities through confidence intervals.

- The formulation using Bayes multiclass error rate (BER), and the adoption of upper bounding BER through global minimal spanning trees is interesting.

- Theorem 5.1 on probability of successful abduction is very insightful. It formalizes the intuition that when the true cause is highly correlated with the observations (i.e., less random), the achievable success rate is high.

Weaknesses:

- The approach is restricted to selective abductive inference where the cause is to be selected from a fixed set. At this point, the problem is akin to classification in the standard ML setting where one has to map from observations to one of the k classes/causes. Is there anyway to extend this approach to "creative" abductive inference?

-  Definition 3.1 says that "Let O denote the vector space with discrete topology containing all binary-featured observation vectors whose components indicate the existence or non-existence of some observed outcome. O contains 2^dim(O) possible observation vectors." What if the observation is actually a continuous valued variable such as temperature?

- "we assume that some finite subset C ⊂ S with cardinality k = |C| has been pre-selected as the finite set of plausible causes assumed to contain the true cause. We further assume C includes a “cause” C_other, whose posterior encapsulates the (likely low) combined probability of all other causes in S occurring. With this, we assume that all causes in C are disjoint and that C contains the one true explanation for observation x." . It is okay to assume that the set of causes are finite and one could also somehow ensure that the causes are disjoint but then one would have to allow presence of multiple concurrent causes - the assumption that there is "one true explanation" for observation from the set of finite disjoint causes appears very restrictive.

- The paper lacks any experimental evaluation to understand the utility and scalability of the proposed approach, and also critically examine the assumptions in the problem formulation.

---

> ### Author Response · Authors · 2024-07-14
> **Response to Reviewer HdgV**
>
> Thank you for your feedback and helping us identify where we can improve our work. Here are our revisions and comments for each of your points:
>
> >The approach is restricted to selective abductive inference where the cause is to be selected from a fixed set. At this point, the problem is akin to classification in the standard ML setting where one has to map from observations to one of the k classes/causes. Is there anyway to extend this approach to "creative" abductive inference?
>
> 1. **Selective vs. Creative abduction:** In our revised draft, we added discussion of extending our approach to creative abduction, namely, representing idea generation as selection over an infinite set of causes (see revisions in Section 7). This is currently reserved for future work since treating $\mathcal{C} \subset \mathcal{S}$ as an infinite set complicates some of our results. For our second set of bounds in particular, we would need to extend the ASF (or a similar framework) to search over an infinite search space – the ASF currently assumes a finite, discrete search space. That said, we could easily extend our earlier set of bounds (section 4) to an infinite set of causes, but we would have to use the computationally expensive theoretical definition of the multi-class Bayes’ error rate. Practical computational methods of bounding BER have not yet been explored for infinite classes.
>
> > What if the observation is actually a continuous valued variable such as temperature?
>
> 2. **Definition of Observation Space for Continuous Observations:** Our current framework supports binary-valued observation vectors, but you could represent a continuous value like temperature as some set of additional features such as ["cold", "warm", "hot"]. In this case, any likelihood distribution over $\mathcal{O}$ would place no probability mass over a contradictory observation vector that is both cold and hot. See revisions in Section 3.1.
>
> >"we assume that some finite subset C ⊂ S with cardinality k = |C| has been pre-selected as the finite set of plausible causes assumed to contain the true cause...With this, we assume that all causes in C are disjoint and that C contains the one true explanation for observation x.". It is okay to assume that the set of causes are finite and one could also somehow ensure that the causes are disjoint but then one would have to allow presence of multiple concurrent causes - the assumption that there is "one true explanation" for observation from the set of finite disjoint causes appears very restrictive.
>
> 3. **Assumption of "One True” Explanation:** Thank you for noticing this error. Section 3 of our submitted draft poorly explained our definition of a cause/cause vector and its distinction with a single event. Our definition of "cause” in the simplex $\mathcal{S}$ is the event representation of a likelihood PMF over the observation space. This means one “cause vector” can be associated with multiple concurrent actual causes, so long as all selected cause vectors in narrower set $\mathcal{C}$ remain disjoint (see a brief added explanation at the end of paragraph 3 of section 3.2). The “one true explanation”  (i.e., what actually caused the observation) can then refer to some set of events, so, in short, our framework does account for multiple concurrent causes. In our initial submission, we mistakenly implied the opposite when trying to explain the disjoint property for events in $\mathcal{C}$, and have since revised our draft to clarify this.
>
> > The paper lacks any experimental evaluation to understand the utility and scalability of the proposed approach, and also critically examine the assumptions in the problem formulation.
>
>
> 4. **Lack of Experiments:** We initially planned this to be a purely theoretical paper, but we believe we can easily add basic example(s) of the framework in action for the first set of bounds in the appendix (which we are currently working on). Regarding the practical applicability of our approach, we only assume a set of boolean observations, a set of possible causes (including causes that represent multiple concurrent events) that are disjoint, and upper and lower q-percentile confidence intervals which, as explained in Section 3.3, can be estimated by known numerical/sampling methods. We also provide practically-applicable techniques for bounding BER (Section 4.4). Paragraph 3 of section 6 provides more explanations of practical applicability. For these reasons, in addition to the length of our current draft, we did not think it would be necessary to provide rigorous experiments for scalability.
>
> > Much of the introductory material in the first 5 pages can be shortened.
>
> 5. **Paper Length:** We have shortened the earlier sections of the draft (particularly sections 2 and 3).
>
> Thanks again for your feedback, please let us know your thoughts.

---

> > ### Comment · Reviewer_HdgV · 2024-07-15
> > **Thank you**
> >
> > Thank you for the response and the revisions made to the paper.

---

### Decision · Action_Editor_H1vX · 2024-08-08

**Recommendation:** Reject

**Comment:**

The work put into the revision performed during the review process was appreciated by all reviewers and seen as a step in the right direction. However, in its current form, all three recommend a rejection, especially since its empirical evaluation is still lacking.

To summarize their reasoning the two main critics are:
1. Insufficient experimental evaluation to back the claims of the paper,
2. Some remaining doubts about the clarity and contribution of the work.

As our main focus at TMLR is on interest over pure novelty, I do not consider the second sufficient for rejection, but the first remains valid. To be relevant to the community the claims and proposals should be accompanied by strong empirical evidence.

The consensus is that this would require a major revision. As such I recommend rejection with the strong encouragement of a resubmission of a major revision, after adding a new set of experiments that better highlight the contribution and provide sufficient evidence for it to be of interest to the community.

**Audience:**

The reviewers agree that the paper has the potential to be of interest to part of the TMLR audience, once it has solved the issues mentioned above.

**Claims And Evidence:**

All reviewers agree that in its current form, the central claims of the paper are not supported by sufficient evidence and require an extensive set of additional experiments.

**Resubmission Of Major Revision:**

The authors may consider submitting a major revision at a later time.